# Myogenic microRNAs as Therapeutic Targets for Skeletal Muscle Mass Wasting in Breast Cancer Models

**DOI:** 10.3390/ijms25126714

**Published:** 2024-06-18

**Authors:** Macarena Artigas-Arias, Rui Curi, Gabriel Nasri Marzuca-Nassr

**Affiliations:** 1Programa de Doctorado en Ciencias Mención Biología Celular y Molecular Aplicada, Universidad de La Frontera, Temuco 4811230, Chile; klga.macarena.artigas@gmail.com; 2Interdisciplinary Post-graduate Program in Health Sciences, Universidade Cruzeiro do Sul, São Paulo 01506-000, Brazil; ruicuri59@gmail.com; 3Departamento de Ciencias de la Rehabilitación, Facultad de Medicina, Universidad de La Frontera, Temuco 4811230, Chile

**Keywords:** microRNAs, myogenic miRNAs, myomirs, skeletal muscle loss, muscle atrophy, breast cancer, protein synthesis, protein degradation

## Abstract

Breast cancer is the type of cancer with the highest prevalence in women worldwide. Skeletal muscle atrophy is an important prognostic factor in women diagnosed with breast cancer. This atrophy stems from disrupted skeletal muscle homeostasis, triggered by diminished anabolic signalling and heightened inflammatory conditions, culminating in an upregulation of skeletal muscle proteolysis gene expression. The importance of delving into research on modulators of skeletal muscle atrophy, such as microRNAs (miRNAs), which play a crucial role in regulating cellular signalling pathways involved in skeletal muscle protein synthesis and degradation, has been recognised. This holds true for conditions of homeostasis as well as pathologies like cancer. However, the determination of specific miRNAs that modulate skeletal muscle atrophy in breast cancer conditions has not yet been explored. In this narrative review, we aim to identify miRNAs that could directly or indirectly influence skeletal muscle atrophy in breast cancer models to gain an updated perspective on potential therapeutic targets that could be modulated through resistance exercise training, aiming to mitigate the loss of skeletal muscle mass in breast cancer patients.

## 1. Introduction

Breast cancer is a major cause of cancer mortality among women worldwide and corresponds to the type of cancer with the highest incidence globally, accounting for 11.7% of all new cases diagnosed in 2020 [1].

Antineoplastic therapeutic options have significantly improved the survival of women diagnosed with this disease. However, antineoplastic treatments such as surgery, chemotherapy, radiotherapy, and endocrine therapy can cause adverse side effects, such as cancer-related fatigue (CRF), cardiotoxicity, decreased muscle strength, functional capacity, and physical performance, and alterations in body composition, which primarily translate into a significant skeletal muscle mass loss or skeletal muscle atrophy, fundamentally limiting the quality of life (QoL) and well-being of these patients [2,3,4].

Skeletal muscle atrophy in breast cancer patients is frequently evaluated using the skeletal muscle index (SMI), measured by computed tomography (CT) at the lumbar vertebra L3. A common criterion for defining significant skeletal muscle loss is having an SMI two standard deviations below the average of healthy young adults, generally less than 5.45 kg/m^2^ in women, according to Prado et al., 2009 [5]. This condition represents a significant challenge for medical care, especially considering its prevalence (~25%) in this population [6]. Although breast cancer does not have the highest incidence of skeletal muscle loss compared to other types of cancer, it is crucial to note that skeletal muscle atrophy is associated with postoperative complications, increased chemotherapy toxicity, and reduced survival in these patients [7]. Zhang XM et al., 2020, carried out a systematic review with meta-analysis and concluded that early-stage breast cancer patients with lower-than-expected skeletal muscle mass for their age had a significantly higher risk of all-cause mortality compared to women with breast cancer without this condition, regardless of age [8].

Skeletal muscle mass can be used as a prognostic factor in patients diagnosed with breast cancer [9,10]. The loss of skeletal muscle mass is considered a significant clinical condition experienced by many oncology patients because of the disease and intensive anticancer therapies, leading to an increase in the inflammatory environment and positive regulation of muscle atrophy genes expressions [11]. The imbalance where catabolic processes outweigh anabolic ones is the underlying cause of skeletal muscle atrophy [12].

Skeletal muscle is a highly dynamic tissue that responds to different stimuli, such as changes in mechanical loading and nutritional intake, that subsequently lead to changes in gene transcription and translation [13,14,15,16]. In addition to well-known transcription factors, noncoding ribonucleic acids (RNAs) have garnered significant interest in the past decade, especially microRNAs (miRNAs). These small molecules, typically comprising 20 to 22 nucleotides, play a pivotal role in modulating transcription pathways and cellular signalling in protein synthesis and degradation. Their influence extends from maintaining skeletal muscle homeostasis to impacting pathological conditions such as cancer [17,18].

Intracellularly, the primary function of a miRNA is to negatively regulate the expression of target genes by interacting with the 3′ untranslated region (3′ UTR) of the target messenger RNA (mRNA) [18]. miRNAs are also released into circulation from cells, encapsulated in extracellular vesicles (EVs) or bound to RNA-binding proteins. These circulating miRNAs also serve as a promising type of biomarker and have been detected in almost all body fluids [19]. More than 2500 miRNAs have been identified in the human genome, where they regulate various protein-coding genes [20].

miRNAs are currently used as biomarkers of the degree of muscle atrophy and are considered therapeutic targets to control the loss of skeletal muscle mass [21,22,23]. The primary therapeutic approach against this loss of muscle mass has been resistance exercise training, also known as strength training. Studies at the molecular level have revealed that 12 weeks of resistance training in older people and 16 weeks of the same type of training in patients with prostate cancer cause changes in expressions of circulating miRNAs involved in myogenesis and the regeneration of skeletal muscle tissue [24,25].

This narrative review aims to identify miRNAs that could directly or indirectly influence skeletal muscle atrophy in breast cancer models to gain an updated perspective on potential therapeutic targets that could be modulated through resistance exercise training to mitigate skeletal muscle mass loss in breast cancer patients.

## 2. Pathways Involved in the Regulation of Skeletal Muscle Mass under Homeostasis

A balance is established between the synthesis and degradation pathways of muscle proteins, which allows for maintaining muscle tropism [22] to maintain homeostasis (Figure 1). The main pathway involved in protein synthesis is Akt/mTOR/S6, which activates insulin-like growth factor I (IGF1) through phosphatidylinositol 3 kinase (PI3K) and captures anabolic signals that induce protein kinase B (Akt) phosphorylation. In turn, Akt phosphorylates the tuberous sclerosis complex 2 (TSC2) protein to inhibit its activity, which results in the activation of mammalian target of rapamycin (mTOR), considered the master regulator of protein synthesis through the action of the mTORC1 complex (TORC1). Consequently, the phosphorylation of proteins found downstream of the TORC1 complex occurs, 70 kDa ribosomal S6 protein kinase (p70S6K) and S6 ribosomal protein (S6), thus initiating messenger RNA translation and promoting muscle fibre size increase [23]. Within the same signalling pathway, Akt also inhibits the phosphorylation of glycogen synthase kinase 3-beta (GSK3-beta), which leads to the disinhibition of eukaryotic initiation factor 2B (eIF2b), thus also favouring the synthesis of protein in skeletal muscle [24].

Paradoxically, the PI3K/Akt signalling pathway is also involved in regulating muscle catabolism [26]. Akt phosphorylation causes the forkhead box O protein (FoxO) to remain sequestered in the cytoplasm, unable to activate the transcription of target genes such as E3 ligases atrogin-1/MAFbx (muscle atrophy F box protein [MAFbx]) and muscle ring 1 (MuRF1), considered master genes of protein degradation [27,28], through the ubiquitin-proteasome system (UPS) [29,30] and autophagy–lysosome system [31].

MuRF1 can also be activated by nuclear factor kappa B (NFkB) through inflammatory cytokines such as tumour necrosis factor (TNF-α), interleukin-1 beta (IL-1β), interleukin-6 (IL-6), and interferon gamma (INFγ) [32]. Increased markers of the degradation pathway will produce a decrease in muscle fibre size [33].

## 3. Skeletal Muscle Mass Loss in Breast Cancer Patients

The dysregulation in homeostasis or an imbalance of protein turnover in women with breast cancer has been addressed in a few clinical studies [34,35,36]. There are two publications from the same research team that demonstrated, through RNA sequencing (RNAseq) analysis in the pectoralis major muscle, a greater expression of genes related to ubiquitin-mediated proteolysis, NADH: ubiquinone oxidoreductase core subunit S8 (Ndufs8), and a decrease in the expression of genes related to insulin receptor substrate 1 (IRS1) protein synthesis in women with breast cancer undergoing chemotherapy [34,35]. The above confirms the idea of altered protein turnover in skeletal muscle atrophy associated with cancer. Mijwel et al. (2018) reported no changes in MuRF1 gene expression (involved in the ubiquitin–proteasome system) after chemotherapy in breast cancer patients [36]. However, it is advisable to analyse this result with caution, considering the late collection of the muscle biopsy in this study, conducted after the onset of chemotherapy. At this point, the cellular processes triggering proteolysis tend to revert to normal expression levels when skeletal muscle atrophy has been established [14,37,38,39].

Finally, Møller et al. (2019), in a study involving 10 participants with various types of cancer, investigated the proteins associated with the signalling pathways responsible for protein turnover in the vastus lateralis muscle [40]. The authors reported a decrease in the content of proteins related to skeletal muscle atrophy (atrogin-1/MAFbx and MuRF1), which suggests a decrease in the activity of the ubiquitin–proteasome system (UPS) and autophagy systems. However, it is important to consider that 9 out of 10 participants included in this study performed a late baseline muscle biopsy, that is, after at least one cycle of chemotherapy with Epirubicin and Doxorubicin [41]. Despite the discrepancies between clinical studies describing skeletal muscle atrophy in women with breast cancer undergoing chemotherapy, preclinical studies in rodents that received Doxorubicin (the most prescribed chemotherapeutic agent in women with breast cancer) showed greater activation of all major proteolytic systems (i.e., calpains, ubiquitin-proteasome pathway, and autophagy) [42]. Preclinical models of cancer-induced skeletal muscle atrophy have described that they depend mainly on the UPS, emphasising that autophagy has a minor role in the present process [43]. A decrease in protein synthesis has been shown in oncological patients with pancreatic or lung cancer in whom reduced PI3K/Akt/mTOR signalling has been demonstrated [44,45]. Also, altered mTOR signalling was reported in mice with breast cancer [46]. These studies confirm that chemotherapeutic agents and breast cancer itself can alter the main protein synthesis pathway in skeletal muscle tissue.

Specifically, chemotherapy regimens used in women with breast cancer, such as Doxorubicin and Taxanes, can cause side effects such as fatigue, diarrhoea, and anorexia, which in turn lead to a reduction in anabolic signals critical to inducing the synthesis of protein [47,48]. A decrease in general physical activity has been observed in people with a cancer diagnosis [49,50], which leads to muscle disuse and phenotypic changes in the distribution of muscle fibre types, going from type I to type II [51].

Both the chemotherapeutic treatment and the tumour itself promote the release of proinflammatory cytokines, such as tumour necrosis factor alpha (TNF-α), tumour necrosis factor-like weak inducer of apoptosis (TWEAK), interleukin-6 (IL-6), interleukin-1β (IL-1β), interleukin-8 (IL-8), and intracellular interferon gamma (INFγ), that promote the activation of NF-κB transcription factor, a regulator of gene expressions associated with skeletal muscle degradation [52,53,54]. Therefore, inflammation plays a key role in developing skeletal muscle atrophy in cancer patients [55].

In summary, the pathophysiology of skeletal muscle mass loss in women with breast cancer has not yet been fully elucidated, but current research has shown that it is due to a combination of multiple factors. One of the factors is related to the aging of the population, where it has been observed that the older the person, the greater the loss of skeletal muscle mass (sarcopenia). Another factor related to skeletal muscle atrophy would be physical inactivity and food deprivation. Finally, antineoplastic treatments, such as mastectomy [56], radiotherapy [57], and especially chemotherapeutic agents like Doxorubicin [42], cause skeletal muscle alterations by triggering an increase in the systemic inflammatory state (e.g., IL-6, TNF-a, IL-1), a decrease in the synthesis pathway (e.g., Akt, p70S6k), and an increase in oxidative stress, leading to mitochondrial dysfunction. This process is closely associated with the activation of signalling pathways for protein degradation (e.g., atrogin-1/MAFbx, MuRF1) in skeletal muscle [58,59]. For these reasons, therapeutic interventions that increase or preserve skeletal muscle mass, improve the response to antineoplastic treatment, and promote earlier recovery after breast tumour resection surgery are necessary [60].

## 4. Role of miRNAs in Maintaining Skeletal Muscle Mass Homeostasis

As previously mentioned in the present review, cancer-related loss of skeletal muscle mass is an essential criterion of morbidity, mortality, and general impairment of quality of life in the oncology population. Therefore, it is essential to identify modulators of the skeletal muscle atrophy process, including messenger RNAs, proteins, and small non-coding RNAs such as microRNAs (miRNAs). miRNAs have been shown to play a critical role in skeletal muscle development and homeostasis [61].

miRNAs negatively regulate gene expression through post-transcriptional mechanisms [62,63]. Their known actions include inhibiting protein translation or enhancing mRNA degradation [64]. Consequently, an increase in a specific miRNA leads to a decrease in the corresponding protein product. Moreover, individual miRNAs do not operate independently or exclusively target a single gene. In fact, miRNAs have multiple genetic targets, and each target can be regulated by multiple miRNAs [65].

Specific muscle-specific miRNAs, known as myomiRs (myogenic miRNAs), have been identified to play a significant role in the development and disease of skeletal muscle [66]. Experimental studies have demonstrated that in mice subjected to mechanical overload, the expression of myomiRs is markedly altered. This suggests that myomiRs have a regulatory role in the hypertrophic process. In this context, reducing the expression of miR-1 and miR-133 during overload leads to positive regulation of IGF-1, thus promoting the muscle protein synthesis pathway [67].

Several miRNAs have been identified that can modulate the IGF-1/Akt/mTOR pathway. The expression levels of miR-1 and miR-133 are reduced during muscle hypertrophy after resistance exercise training with amino acid intake in young men [68]. On the other hand, reducing miR-206 expression levels in skeletal muscle promotes an increase in IGF-1 expression [69,70].

Phosphatidylinositol-3,4,5-trisphosphate 3-phosphatase (PTEN) suppresses Akt activation [71]. In in vivo models, miR-486 increases the phosphorylation status of Akt with a decrease in the levels of PTEN and FoxO1. Therefore, the inhibition of miR-486 activity induces the activation of skeletal muscle atrophy genes [72,73].

These observations reveal that miRNAs regulate various facets of skeletal muscle protein synthesis and degradation pathways, suggesting their potential as therapeutic agents against the loss of skeletal muscle mass. However, the expression profile of miRNAs can vary depending on the type of skeletal muscle atrophy and the specific characteristics of the disease that causes it [74,75].

## 5. Search on Skeletal Muscle Atrophy miRNAs in Breast Cancer

A systematic search was performed in PUBMED and EMBASE using the following search terms: “breast cancer”, “circulating microRNAs”, “microRNA”, “miRNAs”, “skeletal muscle”, “muscle wasting”, “cachexia”, “sarcopenia”, and “Muscular Atrophy” to identify miRNAs deregulated in skeletal muscle atrophy in a breast cancer animal model. Systematic or narrative reviews, in silico studies, and research that evaluated miRNAs unrelated to the regulation of skeletal muscle mass in a breast cancer were excluded.

The search strategy identified 81 studies from the two databases, with 17 duplicate studies. Of the remaining 47 studies, 37 were excluded by reviewing their titles and abstracts. The full text of the remaining 10 articles was reviewed. Ultimately, 5 studies primarily utilising animal models and human breast cancer cell lines [76,77,78,79,80] met the eligibility criteria and were included in the qualitative analysis of the present review (Figure 2 and Table 1).

Of the studies analysed qualitatively, four miRNAs (miR-486, miR-206, miR-122, and miR-155) were identified that were deregulated in skeletal muscle atrophy associated with breast cancer (Table 2). Among the identified miRNAs, miR-486 was the most studied, demonstrating that decreased expression levels in skeletal muscle caused atrophy due to elevated expression of PTEN and FoxO1. This association was consistently observed in three studies that used mouse models of breast cancer [76,77,79]. These results are consistent with previous research which has demonstrated the role of miR-486 in skeletal muscle as a potent modulator of the PI3K/Akt pathway, attenuating PTEN and FoxO1 [81,82]. The reason that miR-486 levels in skeletal muscle and serum decreased in patients with breast cancer compared to healthy individuals is because the breast cancer tumour itself induces a negative regulation of the expression of these microRNAs through exosomes or extracellular vesicles that allow for intercellular communication between tissues [79,83,84,85].

The expression level of miR-206 was upregulated in non-cachectic mice in a breast cancer model [72]. miR-206 was specifically expressed in skeletal muscle, and the inhibition of miR-206 promoted skeletal muscle growth through positive regulation of IGF-1 expression [79]. The elevated levels of the present microRNA observed in muscle atrophy related to breast cancer also generated attenuation of the protein synthesis pathway.

The high expression of miR-122 in animal models of breast cancer was associated with an increase in calpain and UPS activity, ultimately leading to the catabolism of skeletal muscle cells [78]. Other research has also shown that overexpression of miR-122 suppresses the transforming growth factor β (TGF-β)/Smad pathway, which results in the inhibition of myogenesis, a crucial stage in skeletal muscle regeneration [86].

Lastly, the upregulation of miR-155 was found to be a key factor in the degradation of myofibrils and the catabolism of skeletal muscle cells [78,80]. Similarly, other research has shown that the overexpression of miR-155 is a direct result of the increase in proinflammatory factors associated with skeletal muscle atrophy, triggering a decrease in the proliferation and migration of myoblasts and an increase in skeletal muscle apoptosis [86].

Therefore, identifying dysregulated miRNA profiles during skeletal muscle atrophy in breast cancer models enhances our understanding of skeletal muscle biology in diseased conditions. Additionally, it aids in pinpointing relevant therapeutic targets for effective interventions, such as resistance exercise training.

## 6. miRNAs as Potential Therapeutic Targets of Resistance Exercise Training in a Breast Cancer

The clinical benefits of physical exercise have been extensively studied in women with breast cancer, especially in adjuvant stages and in disease survivors not undergoing active cancer treatment [17,87,88,89]. To date, scientific evidence has demonstrated significant clinical improvements, including increased muscle strength, enhanced functional capacity, and reduced fat mass, by implementing individualised concurrent training programs [88,89]. However, resistance exercise training (also called strength training) in isolation has significantly increased muscle size, strength, and power and decreased fatigue in female breast cancer survivors [82]. Resistance exercise training has been studied in female survivors with breast cancer and under adjuvant treatment, proving to be safe and feasible. Resistance exercise training of 12-week duration using machines for lower extremities (3×/week) prior to oncologic surgery has also increased total body muscle mass by 2.72%, muscle strength by 25.5%, and functional capacity by 6.41% (as measured by the distance covered in the 6 min walk test). It has also reduced cancer-related fatigue (CRF) by 28.2% [87]. In women undergoing adjuvant chemotherapy for breast cancer, the experimental group, which engaged in resistance exercise training for 12 weeks using machines (2×/week), exhibited reduced cancer-related fatigue by 7.36%, preserving quality of life compared to the control group. The control group, which did not undergo resistance exercise training, experienced a 44% increase in fatigue and a 7.58% decrease in quality of life [90]. In another study involving breast cancer patients undergoing adjuvant chemotherapy, 21 weeks of resistance exercise training (3×/week) on machines increased lean body mass by 2.91% and overall muscle strength by 9.57% compared to the control group, which had a decreased lean body mass by 1.57%, with no change in muscle strength [91].

Despite the large amount of clinical evidence supporting resistance exercise training in women diagnosed with breast cancer, the effect of this type of training on miRNAs identified as dysregulated during skeletal muscle atrophy in breast cancer patients is unknown. The first study that has investigated the effects of resistance exercise training on miRNAs related to the loss of skeletal muscle mass due to cancer has been carried out in patients with prostate cancer after androgen deprivation therapy. The results of that study have shown that 16 weeks of resistance training increased skeletal muscle mass and strength. Furthermore, there was an increase in the expression level of miRNA-1, miRNA-29, and miR-133 in plasma in the experimental group and a reduction in the expression of the same miRNAs in the control group. These results were contrary to what was expected because the literature described that the decrease in the expression levels of miRNA-1, miRNA-29, and miR-133 is related to the greater activation of the cell growth pathways involved in protein synthesis. However, the increase in miRNA-1, miRNA-29, and miR-133 after resistance exercise training would decrease the activation of the molecular pathways involved in tumour growth (IGF/Akt/mTOR) at a systemic level [25].

These results were similar to those reported by Hagstrom AD et al. in 2018 for breast cancer survivors who had completed antineoplastic treatment. The participants were divided into a control group, receiving usual care, and an experimental group that completed 16 weeks of resistance exercise training. Following the intervention, the levels of expression of circulating miRNAs in serum implicated in skeletal muscle development (miR-1-3p, miR-133a-3p, miR-133b-3p, and miR-486-5p) and miRNAs targeted as oncogenes (miR-370-3p, miR-432-5p, miR-433-3p, miR-493-3p, and miR-654-5p) were compared. There were no statistically significant changes in any of the miRNAs between the groups after resistance exercise training (all *p* > 0.05). Therefore, the authors decided to evaluate the abundance of miRNAs in patients with high and low responses to resistance exercise training. Their findings revealed that patients with a high response demonstrated an upregulation of miR-133a-3p, consistent with previous observations in patients with prostate cancer [92].

These controversial results could stem from the sample obtained to characterise dysregulated miRNAs’ profiles in the cancer-related skeletal muscle atrophy model, since plasma only contains 5% of exosomes containing miRNAs derived from skeletal muscle following a resistance exercise training session [88]. The miRNAs involved in skeletal muscle mass loss are specific in regulating the growth and development of this tissue, but they are also involved in modulating molecular pathways that regulate the maintenance of other cell types and tumour progression [93].

Lastly, the expression levels of exosomes containing miRNAs are influenced by various factors, such as the mode, intensity, and duration of exercise programs. The health status of the participants also plays a role because, as we have seen in the present narrative review, it can strongly influence the profiles of miRNAs present in the sample under study [94,95]. For this reason, it would be reasonable to focus the study of dysregulated miRNAs in cancer-related muscle atrophy on the skeletal muscle tissue involved [96].

## 7. Discussion and Future Directions

In this narrative review, we investigated the mechanisms responsible for skeletal muscle atrophy in breast cancer models, identifying four key miRNAs (miR-486, miR-206, miR-122, and miR-155) that deregulate muscle homeostasis. Figure 3 summarises these miRNAs and their targets within the PI3K/Akt/mTOR and atrogin-1/MAFbx and MuRF1 pathways.

Physical inactivity together with starvation leads to a decrease in the activation of the protein synthesis pathway in people with breast cancer. On the other hand, both oncological pathology and chemotherapeutic agents increase the secretion of inflammatory markers that produce greater activation of the protein degradation pathway, an effect increased by the action of miRNAs. 

Skeletal muscle atrophy in breast cancer patients is multifactorial and cannot be explained by the deregulation of a single molecular mechanism. This fact represents a significant obstacle to preventing a substantial loss of skeletal muscle mass and providing effective therapeutic treatments in this population. However, current studies suggest that resistance exercise training (RET) influences multiple epigenetic factors that modify the expression of genes important for the maintenance and gain of skeletal muscle mass [97].

Research on miRNAs as therapeutic targets of resistance training is still in its early stages. The few findings reported so far include significant shortcomings in study designs, such as small sample sizes, a lack of control groups, and inadequate selection of biological samples, which can be crucial for correct interpretation and reproducibility of the results. Specifically, miRNA analysis represents a challenge given their nature as RNA molecules [98].

Free circulating miRNAs are easily detectable in plasma or other body fluids, allowing for non-invasive identification. However, miRNAs contained in plasma are more exposed to degradation by serum RNases [99,100]. For this reason, the analysis of miRNAs contained in exosomes, microvesicles, apoptotic bodies, and skeletal muscle tissue samples has been encouraged.

The skeletal muscle biopsy (for example, using the Bergström needle) is the only technique that allows for an exhaustive investigation of the cellular mechanisms of cancer-associated skeletal muscle atrophy, including the study of epigenetic alterations, such as the role played by different miRNA profiles in the silencing of key proteins [101]. However, it is an invasive, costly technique and is questioned by ethics committees due to the health condition of these patients, which explains the choice of less invasive samples for the study of these miRNAs [102,103].

Currently, studies on miRNAs implicated in skeletal muscle atrophy in humans diagnosed with breast cancer are limited. A significant limitation of this paper is that the data obtained for this narrative review mainly come from animal models and in vitro cell cultures. Although animal models do not replicate all aspects of muscle atrophy in humans, the results of preclinical studies provide a controlled and valuable model for identifying miRNA profiles involved in the molecular pathways that regulate skeletal muscle mass in breast cancer models. However, those miRNAs that showed promising results in preclinical studies still have a long way to go before being implemented in clinical settings [104].

This study is relevant because it addresses a relatively unexplored area: the role of miRNAs in regulating skeletal muscle mass in breast cancer models. Our work is valuable in oncology as it provides a comprehensive view of how miR-486, miR-206, miR-122, and miR-155 can serve as therapeutic targets to combat cancer-induced skeletal muscle atrophy. Our review advances scientific knowledge by integrating data from multiple studies and highlighting the involved molecular mechanisms. It suggests new directions for future research and the development of clinical interventions, improving breast cancer patients’ quality of life and health outcomes.

## 8. Conclusions

This review underscores the potential of miR-486, miR-206, miR-122, and miR-155 as therapeutic targets to combat skeletal muscle atrophy in breast cancer models. By integrating miRNA research with clinical interventions such as RET, there is an opportunity to mitigate skeletal muscle atrophy and significantly improve outcomes and quality of life for patients with this type of neoplasia. It is imperative to conduct further clinical research to understand these miRNAs therapeutic potential fully.

## Figures and Tables

**Figure 1 ijms-25-06714-f001:**
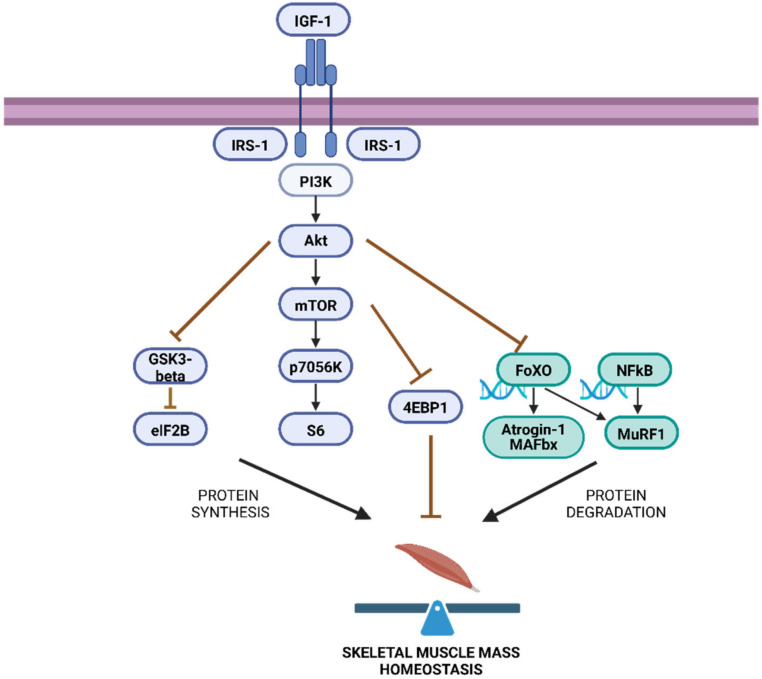
Canonical pathway of skeletal muscle synthesis and degradation. Created with BioRender.com.

**Figure 2 ijms-25-06714-f002:**
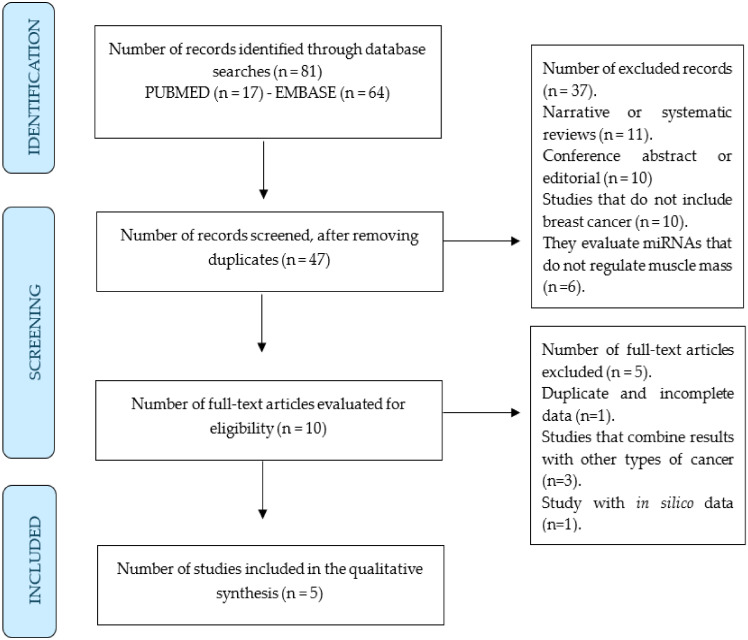
PRISMA flow chart of the search strategy.

**Figure 3 ijms-25-06714-f003:**
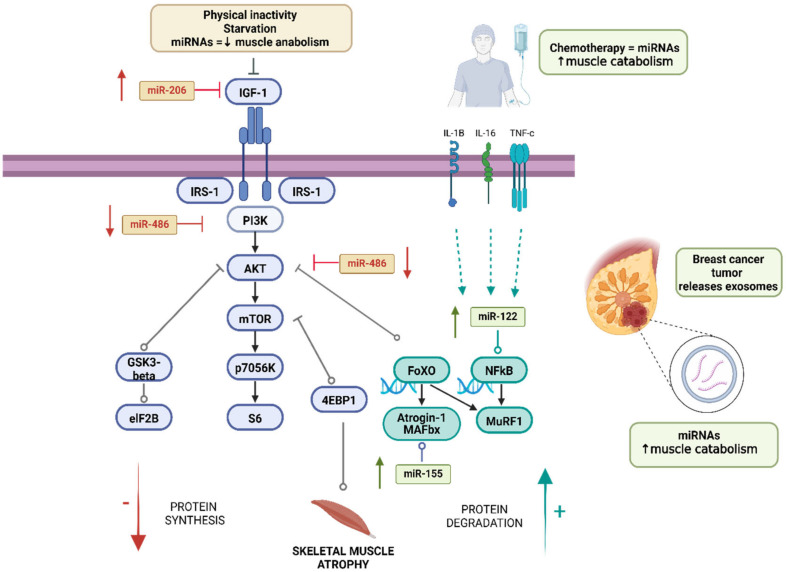
Deregulation of protein synthesis and degradation pathways that regulate skeletal muscle mass in people with breast cancer. Created with BioRender.com.

**Table 1 ijms-25-06714-t001:** The characteristics of the studies included in this narrative review.

Study	Country	Aim	Population	miRNA	Function
Wang R (2022) [76].	United States	To investigate whether the overexpression of miR-486 can overcome skeletal muscle defects in a breast cancer model.	Neu/miR-186+ female transgenic mice representing HER2 breast cancer subtype and miR-186 overexpression.	miRNA-486	It plays an integral role in the myogenesis signalling network involving Pax7, MyoD, myostatin, and NF-κB. It blocks PTEN which upregulates PI3K/Akt.
Chen D (2014) [77].	United States	To identify miRNAs that are present at a lower level in circulation in breast cancer models and their role in skeletal muscle.	Female MMTV-PyMT and MMTV-HER2 transgenic mice representing two subtypes of breast cancer.	miRNA-486	It downregulates PTEN and FOXO1, which consequently activates the PI3K/Akt pathway in cardiac and skeletal muscle.
Yan W (2022) [78].	United States	To determine whether miR-122 secreted by cancer cells suppresses O-GlcNAcylation (OGT) to promote skeletal muscle proteolysis.	Female mice (different types of breast cancer models MDA-MB-231).	miRNA-122	It regulates OGT, which controls OGT of the RYR1 receptor, releasing cytosolic Ca^+2^ in skeletal muscle, which is why it is involved in muscle homeostasis.
Gomes JLP (2021) [79].	Brazil	To investigate the profile of miRNAs that regulate muscle mass in cachectic CT26 mice and non-cachectic MMTV-PyMT mice.	Cachectic female mice with colon cancer (CT26) and non-cachectic female mice with breast cancer (MMTV-PyMT).	miRNA-486	It regulates PTEN mRNA, a key protein that controls the PI3K/Akt/mTOR pathway.
				miRNA-206	Skeletal muscle-specific miRNA directly and inversely regulates the expression of the IGF1 gene.
Wu Q (2019) [80].	China	To identify breast cell-specific miRNAs involved in cancer cachexia.	Human breast cancer cell lines MCF-7 and MDA-MB-231, C2C12, and HEK 293T.	miRNA-155	It modulates PPARγ, whose main function is to regulate the transcription of genes related to lipid and carbohydrate metabolism and inflammation.

Abbreviations. HER2/neu: human epidermal growth factor receptor 2; Pax7: Paired Box 7; MMTV: mouse mammary tumour virus, HER2: human epidermal growth factor receptor 2; PyMT: combined with the polyoma middle T antigen; MyoD: myogenic differentiation protein 1; NF-KB: nuclear factor kB; PTEN: phosphatase and tensin homolog; PI3K: phosphatidylinositol 3-phosphate kinase; AKT: protein kinase B; FOXO: forkhead box O; OGT: O-GlcNAc transferase; MDA-MB-231: human breast cancer cell line from the University of Texas MD Anderson Cancer Center; CT26: cell line derived from mouse colon tumours; RYR1: ryanodine receptor 1; Ca^+2^: calcium; IGF1: insulin-like growth factor-1; MCF-7: human oestrogen-sensitive breast cancer cell line; MDA-MB-231: human triple-negative breast cancer cell line; C2C12: mouse myoblast cell line; HEK293T: cell line in biological research; PPARγ: peroxisome proliferator-activated receptor gamma.

**Table 2 ijms-25-06714-t002:** miRNAs showing differential expression in breast cancer-related muscle atrophy among the studies included in this narrative review.

Author	Sample	Method Used to Identify miRNAs	miRNAs Differentially Expressed in Muscle Atrophy	The Role of Dysregulated miRNAs in Muscle Atrophy
Wang R (2022) [76].	Plasma and tibialis anterior muscle tissue—gastrocnemius of the hind legs of mouse model.	RT-qPCR	↓ miR-486	↓ Levels of p-AKT, p53, and p-LAMA2. ↓ Muscle strength and performance. ↑ Inflammatory cytokines (IL-6, IL-4, TNF-a, and CCL4).
Chen D (2014) [77].	Serum and C2C12 cells from mouse myoblasts.	RT-qPCR Microarray	↓ miR-486	↓ PI3K/AKT pathway signal. ↑ Expression of PTEN and FOXO1 genes.
Yan W (2022) [78].	Breast cancer cells and mouse EDL muscle tissue.	RT-qPCR	↑ miR-122	↑ Muscle fibre degradation. It suppresses the OGT protein and increases RYR1, which favours the cytosolic transport of Ca^+2^ and induces myofibrillar destruction mediated by the UPS system in skeletal muscle.
Gomes JLP (2021) [79].	Mouse tibialis anterior muscle tissue and serum.	RT-qPCR	Cachectic model ↓ miR-486 ↓ miR-206	↓ PI3K and mTOR protein levels.
			Non-cachectic model ↓ miR-486 ↑ miR-206	↓ PI3K, p-AKT, and mTOR protein levels. ↑ miR-206 is associated with ↓ skeletal muscle mass (IGF1 reverse regulation).
Wu Q (2019) [80].	Human breast cancer cells.	RT-qPCR Microarray	↑ miR-155	↓ PPARγ expression alters the energy metabolism of adipocytes and muscle cells. ↑ Adipocyte lipolysis and skeletal muscle cell apoptosis.

Abbreviations. ↑: increase; ↓: decrease; RT-qPCR: real-time reverse transcriptase–polymerase chain reaction; Akt: protein kinase B; LAMA2: phospho-laminin alpha 2; PI3K: phosphatidylinositol 3-phosphate kinase; PTEN: phosphatase and tensin homolog; FOXO: forkhead box O; EDL: extensor digitorum longus muscle; OGT: O-GlcNAc transferase; RYR1: ryanodine receptor 1; Ca^+2^: calcium; UPS: ubiquitin–proteasome; IGF1: insulin-like growth factor-1; mTOR: mammalian target of rapamycin; PPARγ: peroxisome proliferator-activated receptor gamma.

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
