# Peer review of "Myogenic microRNAs as Therapeutic Targets for Skeletal Muscle Mass Wasting in Breast Cancer Models"

_ijms, 2024, doi:10.3390/ijms25126714_

Round 1

Reviewer 1 Report

Comments and Suggestions for Authors

In the review: “MYOGENIC MICRORNAs AS THERAPEUTIC TARGETS FOR SKELETAL MUSCLE MASS WASTING IN  BREAST CANCER MODELS: A NARRATIVE REVIEW”, the authors discussed about the identification of  miRNAs that could influence skeletal muscle atrophy in breast cancer models.

 This  manuscript appears interesting, the authors clearly explain the topic point by point. Anyway, we would like to invite the authors  to clarify some points:

1.       Please check the check punctuation and spaces;

2.       Within the introduction, the authors introduce in general the concept of muscle athropy, in this context.

3.     May be, a very short explanation about miRNAs should be useful for the readers;

4.     More figures may be introduced (e.g. about a biochemical pathway);

5. Page 7, line 166: “Finally, there are antineoplastic drugs that lead to skeletal muscle alterations”; only antineoplastic drugs are available? There are other studies?

6. Are on-going studies having as aim the reduction on specific miRNAs?

7.Schematic representations of the most important considered studies should be useful for the readers to fix principal concepts;

8.       Conclusions are too long and not enough clear, please try to better summarize

Comments on the Quality of English Language

Minor editing and spelling mistakes are present in this manuscript

Author Response

Dear Reviewer,

Thank you very much for the opportunity to review the manuscript and for the suggested comments to improve it. Attached is a Word document with detailed, point-by-point responses to the reviewers' comments, as well as the new version of the manuscript. The modifications in the text are indicated in red for easier identification.

We believe that the manuscript has substantially improved after incorporating the reviewers' suggestions. We look forward to your feedback on whether the new version of the manuscript is suitable for publication.

Sincerely, on behalf of all co-authors,

Gabriel Nasri Marzuca-Nassr
Faculty of Medicine, University of La Frontera, Temuco, Chile. Claro solar 115, Temuco, Chile; Telephone: 56 45 2596713; e-mail: [email protected]

Reviewer 2 Report

Comments and Suggestions for Authors

This paper combed the relevant literature in the past 20 years and summarized the miRNAs involved and their possible targets within the PI3K/Akt/mTOR and atrogin-1/MAFbx and MuRF1 pathways in a breast cancer model, which can provide reference for relevant researchers. However, there are two problems:

1. The refining of existing research results is not comprehensive and profound, and no specific mechanism has been proposed.

2. The author does not give specific suggestions for future research.

Comments on the Quality of English Language

Overall, the information in this review is somewhat limited based on the current limited research results, and the number of readers who will benefit from this review may be limited.

Author Response

(The authors gave the same response as above.)

Reviewer 3 Report

Comments and Suggestions for Authors

Authors present a work addressing: ‚Myogenic MicroRNAs as therapeutic targets for skeletal muscle mass wasting in breast cancer models: A narrative review ’. The aim of this paper was to identify miRNAs that could directly or indirectly influence skeletal muscle atrophy in breast cancer models to gain an updated perspective on potential therapeutic targets that could be modulated through resistance exercise training, targeting to mitigate the loss of skeletal muscle mass in breast cancer patients. The topic of the article is interesting for clinical practice. However, the paper presents  few major and minor issues including:

Major
1. I believe the article is missing a concluding paragraph or short discussion section.
2. Also, authors should strongly emphasize the unique character of the paper, why this review is important?, why their work is valuable in the oncology field? As a separate paragraph.

Minor
1. The article was prepared contrary to the journal's guidelines, including  literature, positioning of tables and figures, font, line spacing.
2.Authors should pay attention to punctuation, lines 66, 112, 124, etc.
3. All abbreviations should be explained as follows: insulin-like growth factor 1 (IGF-1), lines 96, 97, 100, 101, 102, 104, 105, etc.
4. Please explain the meaning of TSC2 protein (line 98).
5. In my opinion authors should add limitation of their literature analysis.

Comments on the Quality of English Language

Modification of the grammar, typos and punctuation is required.

Author Response

(The authors gave the same response as above.)

Round 2

Reviewer 3 Report

Comments and Suggestions for Authors

I recommend to accept the manuscriptin it’s present form.